# Large-scale analysis of small molecule-RNA interactions using multiplexed RNA structure libraries
Ryosuke Nagasawa [1,2,5], Kazumitsu Onizuka [1,2,3,5,6] ✉, Kaoru R. Komatsu [4,5], Emi Miyashita[4], Hirotaka Murase [1], Kanna Ojima[1,2], Shunya Ishikawa[1,2], Mamiko Ozawa[1], Hirohide Saito [4,6] ✉ & Fumi Nagatsugi [1,2,6] ✉

The large-scale analysis of small-molecule binding to diverse RNA structures is key to understanding the required interaction properties and selectivity for developing RNA-binding molecules toward RNA-targeted therapies. Here, we report a new system for performing the large-scale analysis of small molecule–RNA interactions using a multiplexed pull-down assay with RNA structure libraries. The system profiled the RNA-binding landscapes of G-clamp and thiazole orange derivatives, which recognizes an unpaired guanine base and are good probes for fluorescent indicator displacement (FID) assays, respectively. We discuss the binding preferences of these molecules based on their large-scale affinity profiles. In addition, we selected combinations of fluorescent indicators and different ranks of RNA based on the information and screened for RNA-binding molecules using FID. RNAs with high- and intermediate-rank RNA provided reliable results. Our system provides fundamental information about small molecule–RNA interactions and facilitates the discovery of novel RNA-binding molecules.

Targeting RNA with small molecules represents an attractive medicinal approach for treating gene-related and infectious diseases[1–5]. For example, drugs targeting specific RNA splice sites have been approved to alleviate the symptoms of spinal muscular atrophy[6,7]. Further, human precursor microRNAs (pre-miRNAs)[8–13], various repetitive RNAs, such as CUG[14–17] and UGGAA[18] repeats, and structured RNA elements of infectious pathogens[19–21] are considered promising drug targets. When developing new RNA-binding molecules, profiling the small molecule-binding landscapes of various types of RNA structures is critical for gaining deep insights into their binding properties and selectivities[22–24]. One powerful way to profile the binding of small molecules is an analysis based on massively parallel DNA sequencing. For example, Disney's group developed a computational approach, Inforna, based on their screening methods and massive sequencing analysis, that has led to the discovery of various regulatory RNA-binding molecules in RNA-related disease models[10–12,25]. Their binding profiles focused on the sequence variants within internal loops and bulge structures. More recently, Sugimoto's group implemented RNA-capturing

microsphere particles to establish a new sequencing-based RNA-selection method that does not require any ligand labeling for the RNA-binding fluorescent molecules[26,27]. Although these methods are valuable, they could produce inaccurate results in the profiling of specific or stable RNA structures, such as G-quadruplex (G4) structures, owing to structure-dependent amplification biases. This is because polymerase tends to pause at structured RNA sites during reverse transcription or polymerase chain reactions (PCR)[28,29]. Therefore, different approaches that do not involve reverse transcription or PCR are required for the profiling of small-molecule binding to diverse RNA structures, particularly highly structured RNAs exhibiting naturally occurring sequences.

Recently, we developed a new method, folded RNA element profiling with structure library (FOREST)[30], for the large-scale analysis of protein–RNA interactions using a multiplexed RNA structure library. FOREST quantifies interactions using a DNA barcode microarray that can capture RNA probes in an RNA structure library (Fig. 1) that is designed by extracting structured motifs from RNA structure datasets. In this system, a

[1]Institute of Multidisciplinary Research for Advanced Materials, Tohoku University, Miyagi 980-8577, Japan. [2]Department of Chemistry, Graduate School of Science, Tohoku University, Miyagi 980-8578, Japan. [3]Division for the Establishment of Frontier Sciences of Organization for Advanced Studies, Tohoku University, Miyagi 980-8577, Japan. [4]Center for iPS Cell Research and Application (CiRA), Kyoto University, Kyoto 606-8507, Japan. [5]These authors contributed equally: Ryosuke Nagasawa, Kazumitsu Onizuka, Kaoru R. Komatsu.[6]These authors jointly supervised this work: Kazumitsu Onizuka, Hirohide Saito, Fumi Nagatsugi. ✉e-mail: onizuka@tohoku.ac.jp; saitou.hirohide.8a@kyoto-u.ac.jp; nagatugi@tohoku.ac.jp

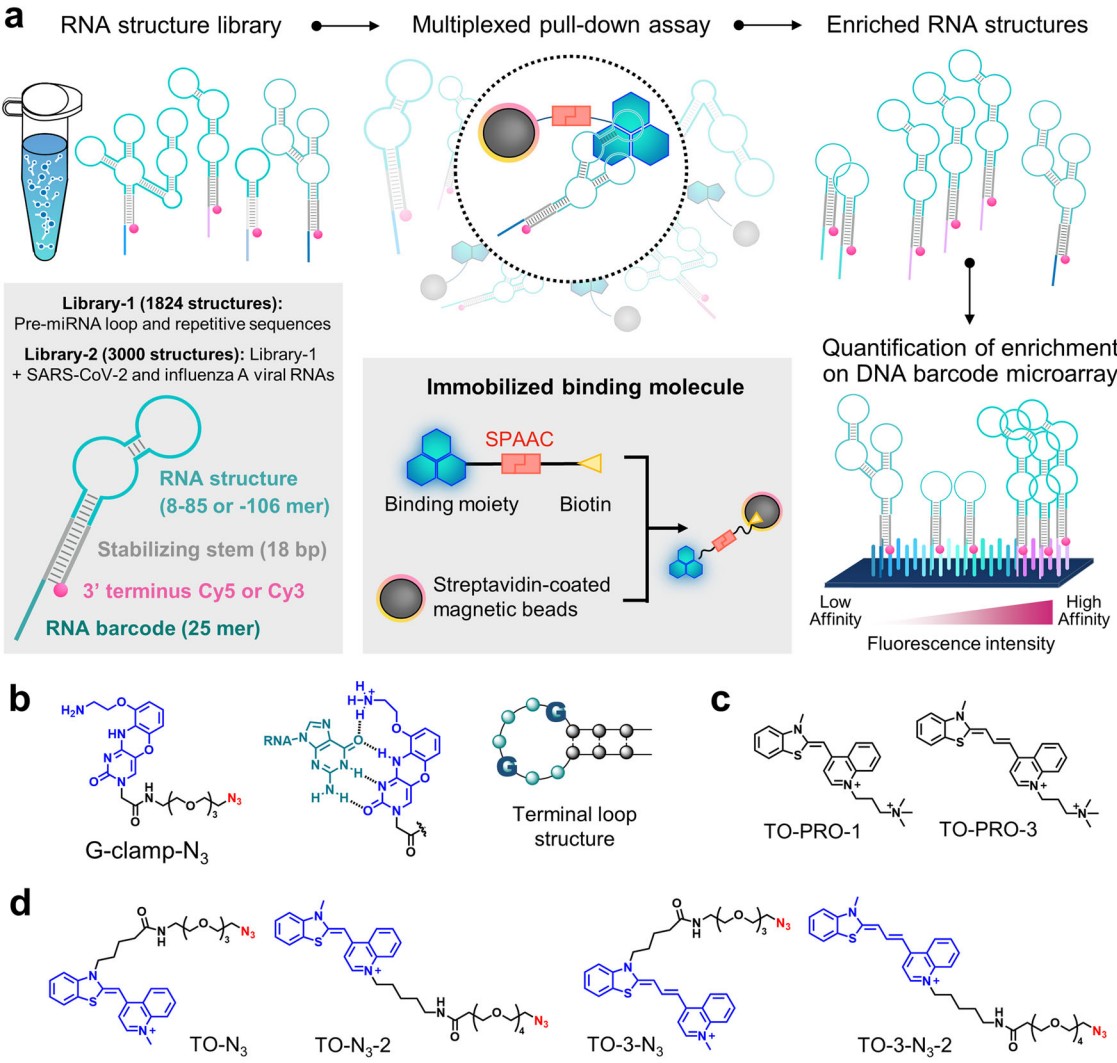

**Fig. 1 | Method overview and the tested small molecules. a** Schematic of the large-scale analysis of small molecule–RNA interactions. The RNA structured library consists of an RNA structure region, a common stabilizing stem region, and a barcode region. The 3' end is modified with a fluorescent group. The RNA structure region has 1824 kinds of structure consisting of pre-miRNA loops and repetitive sequences (Library-1). Library-2 contains library-1 plus SARS-CoV-2 and influenza A viral RNAs. The designed RNA structure library was used for the multiplexed pull-down assay with a small molecule immobilized on streptavidin-coated magnetic beads. The enriched RNA structures were analyzed based on the differences in fluorescence intensity observed on DNA barcode microarrays, and the interactions between small molecules and RNA were quantified. **b** Structure and RNA recognition mode of G-clamp-$N_3$. The binding moiety is shown in blue, the linker is shown in black, and azide is shown in red. **c** Structures of TO-PRO-1 and TO-PRO-3. **d** Structures of TO–$N_3$, TO–$N_3$-2, TO-3–$N_3$, and TO-3–$N_3$-2.

stabilizing common stem, a unique RNA barcode (5' terminus), and Cy5 or Cy3 (3' terminus) were attached to each RNA structure (Fig. 1a). Employing this system, we revealed the interaction landscape of RNA-binding proteins (RBPs) using the RNA structure library that was extracted from human pre-miRNAs, human 5' UTRs, and the HIV-1 RNA genome. FOREST drives amplification-free quantification, thus facilitating the bias-free detection of different RNA structures and their interactors (e.g., G4 and G4-binding RBPs). Notably, we identified cross-reactive interactions among some of the tested RBPs. For example, we observed that three G4-binding proteins exhibited different binding preferences to G4 and interacted with non-G4 RNA motifs (e.g., the r(GAA)$_n$ motif) with different selectivity. Thus, we hypothesized that our method could be used as a platform for profiling the RNA-binding landscapes of small molecules.

In this study, we introduced a systematic and large-scale approach for investigating small molecule–RNA interaction profiles. By subjecting small molecules to FOREST, our system is advantageous for analyzing large-scale datasets of diverse RNA structures derived from naturally occurring sequences. As the detection of the binding affinities of different RNA structures is based on microarray analysis, FOREST avoids sequencing and

structure-dependent amplification biases. Additionally, the results include not only high-affinity interactions but intermediate- and low-affinity ones. Therefore, our datasets will be invaluable resources for understanding the fine determinants of small molecule–RNA interactions.

## Results and discussion
### Design of the platform for the large-scale analysis of small molecule–RNA interactions
Regarding the first RNA structure library for the analysis (Library-1), we designed 1824 RNA structural motifs by extracting the terminal loops of human pre-miRNAs and adding several repetitive and control sequences[30]. Five different barcodes were allocated to each motif structure to exclude the outliers representing non-specific binding to the barcode sequences. Thereafter, the small molecule was immobilized onto beads via biotin–streptavidin interactions (Fig. 1a). We performed the pull-down process by mixing the RNA structure library and immobilizing the small molecule, followed by the washing and elution steps to collect the bound RNAs. The RNAs that were pulled down were quantified by a DNA barcode microarray to obtain the fluorescence intensity of each RNA structure because of the correlation of

**Fig. 2 | Box plots of the number of bases in single-stranded RNA (ssRNA) and double-stranded RNA (dsRNA), as determined by RNA secondary structure prediction.** The boxes were generated for each of the five subpopulations (each comprises 360 RNA structures) based on their rankings, as sorted using the G-clamp binding intensity and overall population (1800 pre-miRNA structures). The box plot elements are defined as follows: center line, median; box limits, upper and lower quartiles; points, outliers. The *p*-values were determined by the two-tailed Brunner–Munzel test with a Bonferroni correction. n.d. means no significant difference.

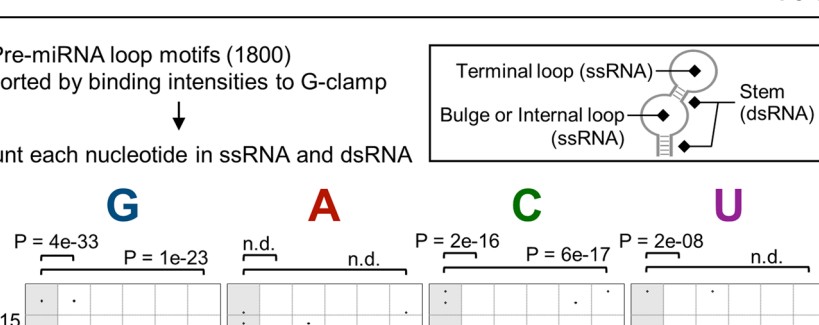

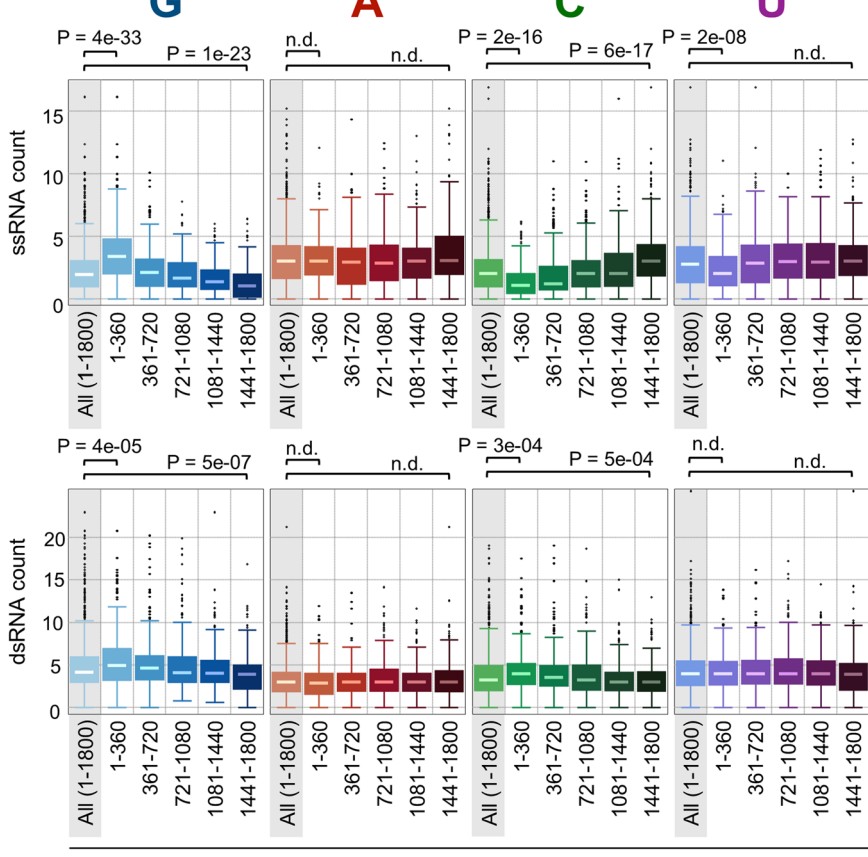

fluorescence intensities with binding affinities after background subtraction by no-ligand-conjugated streptavidin control samples[30].

In this study, we selected G-clamp and thiazole orange (TO) derivatives as the binding molecules (Fig. 1). G-clamp can recognize an unpaired guanine base in RNA loop structures by forming four hydrogen bonds (Fig. 1b)[31–33]. G-clamp was used to validate our system because it binds strongly to a wide range of RNAs. Conversely, the TO derivatives, TO-PRO-1 and TO-PRO-3, are known as fluorescent light-up probes for imaging and fluorescent indicator displacement (FID) assays (Fig. 1c)[34–38]. FID represents a high-throughput method for identifying novel RNA-binding molecules[39–45]. For example, TO-PRO-3, a deep-red fluorescent indicator, was used in an FID assay to screen for compounds that bind to the bacterial A-site, influenza A virus RNA, and G4 DNA[37,38,46]. However, the binding information of these fluorescent indicators and their target RNA sequences is still limited. We believed that it would be beneficial to determine the RNA binding profiles of such conventionally used indicators to further expand the repertoire of target RNA sequences that can be used in FID assays. Based on the structure of TO-PRO-1, we designed the $N_3$-modified TO–$N_3$ and TO–$N_3$-2 exhibiting different linker positions (Fig. 1d). Similarly, we designed TO-3–$N_3$ and TO-3–$N_3$-2. These $N_3$-modified molecules were conjugated to biotin via a strain-promoted azide–alkyne cycloaddition (SPAAC) with DBCO–biotin (Figs. 1a, S1, and S2)[47,48] and used for the large-scale analysis.

**Large-scale analysis of the interaction of G-clamp-$N_3$ with Library-1**

First, we ranked the RNA motifs from Library-1 based on their G-clamp binding (Supplementary Data 1). In Supplementary Data 1, the

sequences, binding scores, Z-scores, and CVs are shown in order of rank. To understand the binding properties of G-clamp, the numbers of bases in the single-stranded (ss) and double-stranded (ds) RNA regions were investigated using the secondary structures of the pre-miRNA loops predicted by RNAsubopt in the ViennaRNA package[49] (Fig. 2). The ssRNA region refers to the terminal loop, bulge, or internal loop. Boxes were generated for each of the five subpopulations based on their rankings. Regarding ssRNA, the G count of high-ranking RNAs (1–360) was significantly higher than that of all the pre-miRNAs in Library-1. Contrarily, the G count of the low-ranking RNAs (1441–1800) was significantly lower than that of all the examined pre-miRNAs. Conversely, the C counts of the high- and low-ranking RNAs were lower and higher than those of all the pre-miRNAs in Library-1, respectively. The U count of the high-ranking RNAs was lower than that of all the pre-miRNAs, and the A count of ssRNA was not significantly different among the rank sections. Regarding dsRNA, the four bases exhibited smaller differences among the ranks compared with ssRNA. The C and U counts were inversely proportional to the G count, as C and U in the ssRNA region can form base pairs with the neighboring G bases. Furthermore, the percentage of the unpaired G count highlighted an unpaired-G selectivity (Fig. S3). Five or more unpaired Gs were mainly observed in high-ranking RNAs (1–180), and the percentage decreased gradually as the rank decreased. Contrarily, few RNAs without any or only a single unpaired Gs were observed in the high-ranking group, and the percentage gradually increased as the rank decreased. These results corresponded to the fact that G-clamp mostly recognizes G base in the ssRNA regions[32].

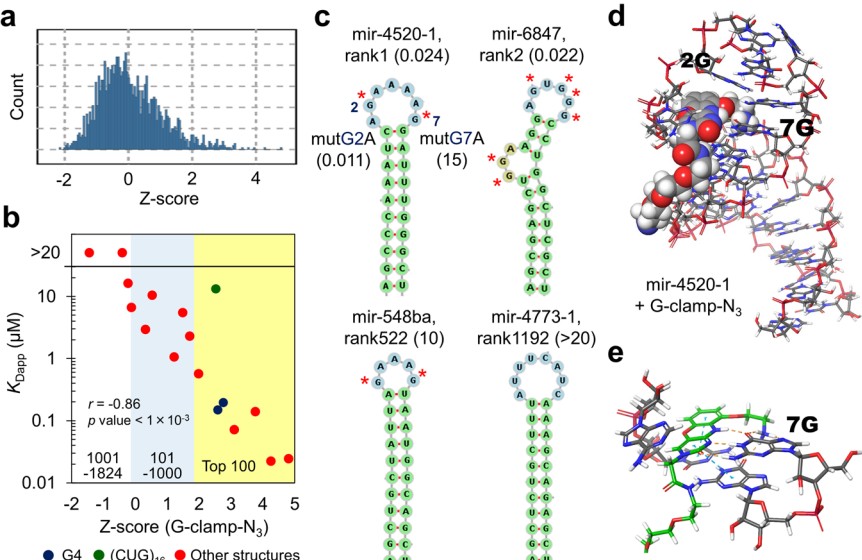

**Fig. 3 | Large-scale analysis of the interaction of G-clamp–N₃ with Library-1 (1824 different sequences). a** Histogram of the $Z$-scores (binding intensities between RNA and G-clamp–N₃). **b** Correlation between the $Z$-scores and apparent dissociation constants ($K_{Dapp}$). The dark blue, green, and red circles indicate G4 RNAs, (CUG)₁₆, and other structures, respectively. The mean data from two independent experiments are shown. $r$ indicates Spearman's correlation coefficient and the $p$-value was determined by a no-correlation test. **c** Representative minimum free energy structures of the pre-miRNA loop motifs in the library. Numbers in

parentheses indicate the $K_{Dapp}$ values (μM). The red asterisks (*) indicate the G base in the single-stranded (ss) regions. **d** Molecular modeling of the complex structure between hsa-mir-4520-1 and G-clamp–N₃. G-clamp–N₃ binds to 7G. The CPK models indicate G-clamp–N₃. The orange dash and blue lines indicate the hydrogen bond and stacking interaction, respectively. The complex structure was modeled by RNAComposer and MacroModel. **e** Modeling structure focusing on interactions with G-clamp. The green molecule indicates G-clamp–N₃.

Next, to validate our screening platform for RNA structures, we selected 17 sequences from the high-affinity (top 100), intermediate-affinity (101–1000), and low-affinity (1001–1824) groups and measured their apparent dissociation constants ($K_{Dapp}$) by fluorescence titration (Fig. S4). To shorten the common stem and keep the RNA motif structures stable in the titration assays, a shorter common stem (three base pairs) was attached to the motifs (5′-AGC-motif-GCU-3′). A histogram of $Z$-scores and the correlation between the $Z$-scores and $K_{Dapp}$ values are shown in Fig. 3a and b and Table S1. The minimum free energy structures of the selected RNAs are shown in Figs. 3c and S5. The ranks 1 and 2 RNAs (Fig. 3c, top) contained unpaired guanine bases in their loop structures and exhibited strong G-clamp binding ($K_{Dapp} = 0.024$ and $0.022$ μM, respectively). For the rank 1 RNA (hsa-mir-4520-1 loop), we performed the G mutation assay using two G-mutated hsa-mir-4520-1 loops (hsa-mir-4520-1-mutG2A and -mutG7A). Although mutG2A exhibited strong binding ($K_{Dapp} = 0.011$ μM) similar to the wild type, mutG7A exhibited weaker binding ($K_{Dapp} = 15$ μM). The double mutant mutG2,7A also exhibited weaker binding ($K_{Dapp} = 3.7$ μM) than the wild type, indicating that G7 contributes to the strong interaction with G-clamp. Surface plasmon resonance (SPR) analysis also showed the same binding tendency as the values obtained by fluorescence titration experiments, although the values slightly increased (Fig. S6). While the wild-type and mutG2A exhibited strong binding ($K_{Dapp} = 0.10 \pm 0.02$ and $0.044 \pm 0.008$ μM, respectively), mutG7A exhibited much weaker binding ($K_{Dapp} > 50$ μM). To consider the selectivity of G7, the molecular modeling of the complex structure between hsa-mir-4520-1 and G-clamp–N₃ was performed using RNAComposer[50,51] and MacroModel (Fig. 3d). When G-clamp is bound to 7G by hydrogen bonds, it can interact with neighboring bases. We considered that these interactions, such as stacking with CG base pair and a hydrogen bond with G base at the top of the stem (Fig. 3e), would facilitate strong binding in addition to the formation of the hydrogen bonds with the target G base. When G-clamp was bound to 2 G by hydrogen bonding, stacking interactions were not observed with neighboring bases (Fig. S7). These results indicate that G-clamp does not recognize all Gs on the loop (G-clamp recognizes specific Gs). The high number of G bases in the ssRNA region of

high-ranking RNAs probably increased the probability of the presence of G bases that bind to G-clamp strongly (Fig. S3). In the high-affinity group, two of the selected RNA motifs contained the G4 structure. The $K_{Dapp}$ values of the hsa-mir-6850 loop (rank 28) and G4_(GGGU)₆ (rank 38) were 0.19 and 0.15 μM, respectively. This may be because G-clamp intercalated on G4 RNAs. In the intermediate-affinity group, even though hsa-mir-548ba (rank 522) exhibited a loop that was similar to that in hsa-mir-4520-1, its $K_{Dapp}$ value (10 μM) was much higher. Comparing the modeling structures of hsa-mir-4520-1 and hsa-mir-548ba (Fig. S8) revealed that G-clamp–N₃ cannot strongly interact with adjacent bases when it forms hydrogen bonds with a G base on the loop structure of hsa-mir-548ba. In the low-affinity group, the loops without any G bases, such as hsa-mir-4773-1 (rank 1192), hsa-mir-4282 (rank 1775), exhibited weak binding ($K_{Dapp} > 20$ μM) and common stem sequence with four Us in the terminal loop also exhibited weak binding ($K_{Dapp} = 9$ μM) (Figs. S4 and S5). Within the group of selected RNAs, only (CUG)₁₆ (rank 43) deviated from our expectations in the fluorescence titration experiment (Fig. 3b, green color). Overall, we observed a good correlation between the $Z$-scores and observed $K_{Dapp}$ (Fig. 3b, Spearman's correlation coefficient: −0.86); the coefficient without considering (CUG)₁₆ exhibited an even higher correlation (−0.95). The G4 structures, which are susceptible to bias when using sequencing-based methods, were evaluated and ranked. These results indicate that our system for the large-scale analysis of the RNA structure libraries can ensure accurate assessments of small molecule–RNA interactions.

## Large-scale analysis of the interaction of the thiazole orange derivatives with Library-2

Next, we investigated the binding of different RNA motifs to the TO derivatives using our second RNA structure library, Library-2 (Supplementary Data 2–5). Library-2 contains 3000 RNA structural motifs that were designed by extracting the terminal loops of human pre-miRNAs, along with SARS-CoV-2 and influenza A virus RNAs and several repetitive and control sequences. Compared with the G-clamp binding profile, TO and TO-3 exhibited distinct profiles (Fig. 4a), although a significant correlation was observed between their binding profiles (Fig. 4b). These data indicate that

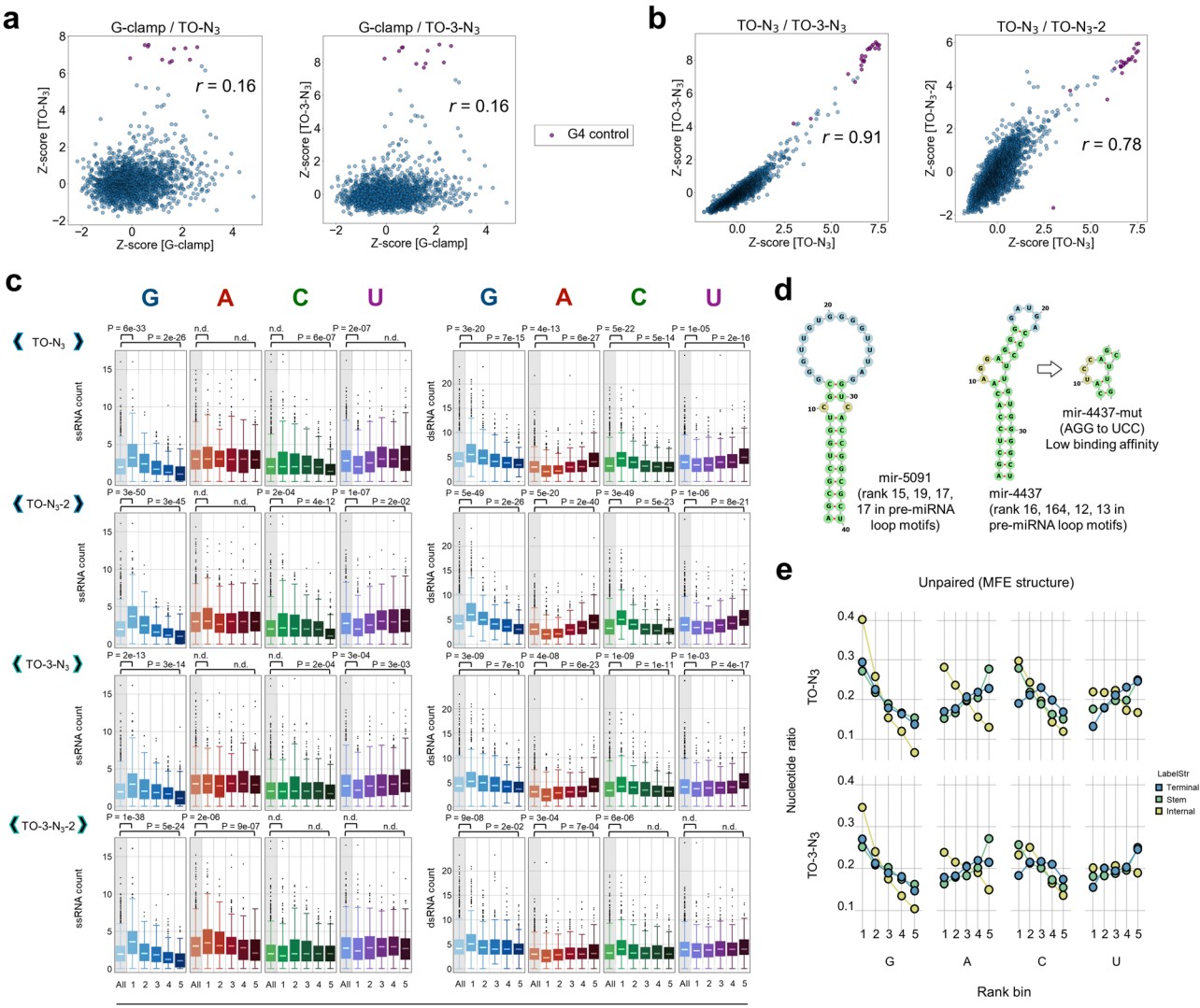

**Fig. 4 | Analysis of the binding properties of the TO derivatives.** *Z*-score correlations of **a** G-clamp/TO–N$_3$ (left) and G-clamp/TO-3–N$_3$ (right), **b** TO–N$_3$/TO-3–N$_3$ (left) and TO–N$_3$/TO–N$_3$-2 (right). The purple circles indicate the G4 control RNAs, and *r* indicates Spearman's correlation coefficient. **c** Box plots of the numbers of bases in ssRNA and dsRNA, as determined by RNA secondary structure prediction. The boxes were generated for each of the five subpopulations (each comprising 360 pre-miRNA structures) based on their rankings, as sorted by the TO or TO-3 binding intensities and overall population (1800 pre-miRNA structures). The *p*-values were determined by the two-tailed Brunner–Munzel test with a Bonferroni correction. n.d. indicates that there was no significant difference. **d** Representative minimum free energy structures in the high-rank RNAs. The four numbers in the parentheses indicate the ranks of TO–N$_3$, TO–N$_3$-2, TO-3–N$_3$, and TO-3–N$_3$-2 in the pre-miRNA loop motifs, respectively. **e** Ratios of the four nucleotides (G, A, C, and U) in the structural motifs, as determined by RNA secondary structure prediction. The data points were generated for each of the five subpopulations (each comprising 360 RNA structures) based on their rankings sorted by the TO or TO-3 binding intensities. The colors represent the structural motifs (terminal loop: blue, stem: green, internal loop: yellow) used for counting the number of nucleotides.

that the TO derivatives exhibited similar selectivities, which were unique compared with the G-clamp, as expected. The correlation coefficient between TO–N$_3$ and TO–N$_3$-2 with different linker positions ($r = 0.78$) was lower than that between TO–N$_3$ and TO-3–N$_3$ with the same linker position ($r = 0.91$), suggesting that the linker positions affect the binding profile (Fig. 4b). The high-affinity group of RNAs for the TO derivatives was mainly populated with G4 RNAs. The kernel density estimation of the *Z*-scores of the TO derivatives indicated the significant enrichment of the G4 control RNAs (Fig. S9).

To understand the binding properties of the TO derivatives, the numbers of bases in the ssRNA and dsRNA regions were quantified using the predicted secondary structure of the pre-miRNA loops similar to the analysis of the G-clamp (Fig. 4c). For ssRNA, the G count of the high-ranking RNAs (1–360) was significantly higher than that of all the pre-miRNAs in Library-2. Contrarily, the ssRNA counts of the other bases were not significantly different among the different ranks. Regarding

dsRNA, the G and C counts of the high-ranking RNAs (1–360), as well as the A and U counts of the low-ranking RNAs (1441–1800), were significantly higher than that of all the pre-miRNAs. The count tendencies of TO-3–N$_3$ and TO–N$_3$ were similar. Overall, these results altogether suggest that the TO derivatives prefer G-rich ssRNA and G/C-rich rigid stem structures, such as hsa-mir-5091 and -4437 (Fig. 4d). Regarding ssRNA, we further examined the total number of nucleotides in the internal and terminal loops (Fig. 4e). Although high-ranking RNAs exhibited more G and A bases in their internal loops, the terminal loops of high-ranking RNAs only exhibited a preference for more G but no other bases. These results suggest that the TO derivatives prefer the G/A bases in the internal and G-rich terminal loops. A likely explanation is that the internal loops comprising G/A bases may create a binding pocket that is ideal for intercalation, whereas the G-rich terminal loops may form G4-like structures. To confirm the preference of the TO derivatives for internal loops comprising G/A bases, we compared the $K_{Dapp}$ values of

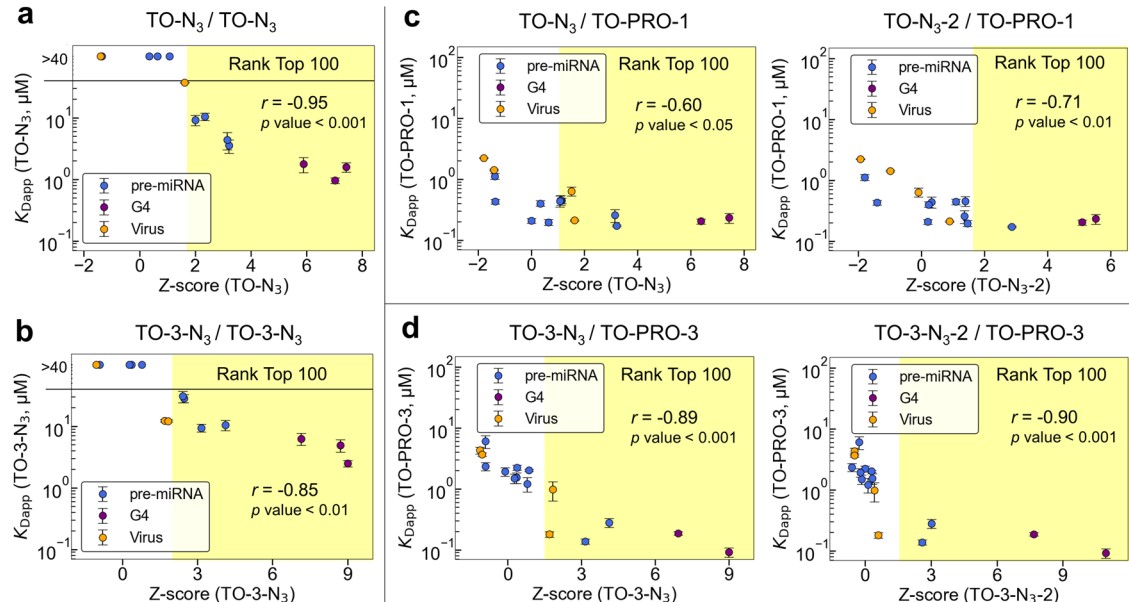

**Fig. 5 | Correlation between the Z-score in FOREST and $K_{Dapp}$ value of the TO derivatives.** The blue circles indicate pre-miRNAs. The purple and orange circles indicate G4 RNAs and virus RNAs, respectively; $r$ indicates Spearman's correlation coefficient; and the p-values were determined by the no-correlation test. The mean data of the three independent experiments are shown. The error bars indicate the standard errors. **a** Z-score (TO–$N_3$) and $K_{Dapp}$ value (TO–$N_3$). **b** Z-score (TO-3–$N_3$) and $K_{Dapp}$ value (TO-3–$N_3$). **c** Left: Z-score (TO–$N_3$) and $K_{Dapp}$ value (TO–PRO-1). Right: Z-score (TO–$N_3$-2) and $K_{Dapp}$ value (TO–PRO-1). **d** Left: Z-score (TO-3–$N_3$) and $K_{Dapp}$ value (TO–PRO-3). Right: Z-score (TO-3–$N_3$-2) and $K_{Dapp}$ value (TO-PRO-3).

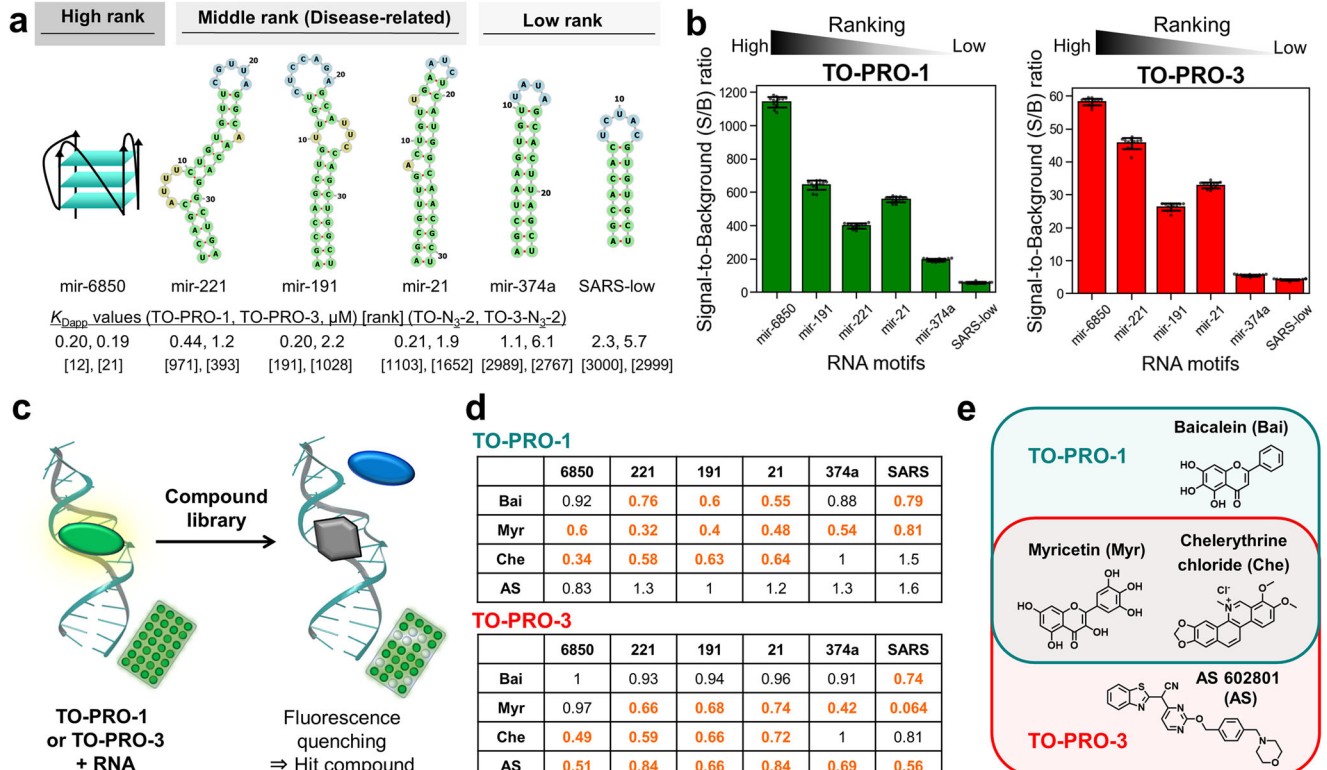

**Fig. 6 | Fluorescent indicator displacement (FID) assay using TO-PRO-1 or TO-PRO-3. a** RNA secondary structures and $K_{Dapp}$ values of TO-PRO-1 and TO-PRO-3, and ranks of TO–$N_3$-2 and TO-3–$N_3$-2 for the target RNA motifs in the FID assays: has-mir-6850 from the high rank (top 100); hsa-mir-221, -191, and -21 from the intermediate-affinity-ranked disease-related human pre-miRNAs (101–1800); hsa-mir-374a and SARS-low from the low rank (>1800). **b** Signal-to-background (S/B) ratios of TO-PRO-1 (left) and TO-PRO-3 (right). Data are mean ± SD (n = 12). **c** Schematic of the FID assay. The fluorescence signal of the RNA-bound and light-up TOs quenches when a compound binds to a target RNA at the same site. **d** Summary of the FID assays. The numbers indicate the normalized fluorescence of the hit compounds in the FID assay using TO-PRO-1 or TO-PRO-3. Orange and bold numbers indicate the hit combinations of RNA and hit compound. **e** Structures of the hit compounds.

**Fig. 7 | Validation of fluorescent indicator displacement (FID) assay results. a** $K_{Dapp}$ values (μM) of the hit compounds. The $K_{Dapp}$ values were measured by fluorescence titration. The mean data of the three independent experiments are shown. The errors indicate the standard errors. **b** Summary of the true–false results for the FID assay using TO-PRO-1. Compounds with $K_{Dapp}$ values ≤ 40 are defined as those bound to the target RNAs. Compounds with $K_{Dapp}$ values > 40 are defined as those unbound to the target RNAs. True indicates that the FID assay data are consistent with the $K_{Dapp}$ value (hit compounds/bound compounds or non-hit compounds/unbound compounds). False indicates that the FID assay data are not consistent with the $K_{Dapp}$ value (hit compounds/unbound compounds or non-hit compounds/bound compounds). N and P indicate negative and positive results, respectively. **c** Summary of true–false results for the FID assay using TO-PRO-3.

**a** $K_{Dapp}$ values (μM)

| | High rank | Middle rank (Disease-related) | | | Low rank | |
|---|---|---|---|---|---|---|
| | mir-6850 | mir-221 | mir-191 | mir-21 | mir-374a | SARS-low |
| Bai | >40 | >40 | >40 | >40 | >40 | >40 |
| Myr | 3.3 ± 0.1 | 20 ± 2 | 25 ± 3 | 16 ± 0.5 | >40 | >40 |
| Che | 4.0 ± 0.5 | 3.8 ± 0.3 | 3.8 ± 0.5 | 2.6 ± 0.6 | 12 ± 1 | 16 ± 0.7 |
| AS | 4.5 ± 0.4 | >40 | 14 ± 2 | 20 ± 0.9 | >40 | >40 |

**b** TO-PRO-1

| | High rank | Middle rank (Disease-related) | | | Low rank | |
|---|---|---|---|---|---|---|
| | mir-6850 | mir-221 | mir-191 | mir-21 | mir-374a | SARS-low |
| Bai | True N | False P | False P | False P | True N | False P |
| Myr | True P | True P | True P | True P | False P | False P |
| Che | True P | True P | True P | True P | False N | False N |
| AS | False N | True N | False N | False N | True N | True N |

**c** TO-PRO-3

| | High rank | Middle rank (Disease-related) | | | Low rank | |
|---|---|---|---|---|---|---|
| | mir-6850 | mir-221 | mir-191 | mir-21 | mir-374a | SARS-low |
| Bai | True N | True N | True N | True N | True N | False P |
| Myr | False N | True P | True P | True P | False P | False P |
| Che | True P | True P | True P | True P | False N | False N |
| AS | True P | False P | True P | True P | False P | False P |

hsa-mir-4437 and its internal loop (AGG to UCC) mutant, mir-4437-mut (Figs. 4d and S10). Although the $K_{Dapp}$ values of TO–N$_3$ and TO-3–N$_3$ for the wild type hsa-mir-4437 loop were relatively low, 4.4 and 11 μM, respectively, the $K_{Dapp}$ values of mir-4437-mut were much higher (>40 μM), suggesting that the G/A bases in the internal loop are crucial to the strong binding of the TO derivatives to the hsa-mir-4437 loop at least.

To further validate the binding profiles of the TO derivatives that were generated by our screening platform, the $K_{Dapp}$ values of TO–N$_3$ and TO-3–N$_3$ interacting with 15 RNAs (pre-miRNAs, G4 RNAs, and virus RNAs) were measured by fluorescence titration (Figs. S11 and S12 and Table S2). For the high-ranking RNAs (top 100), the $K_{Dapp}$ values correlated well with the Z-scores of TO–N$_3$ and the Spearman correlation coefficient was −0.95 (Fig. 5a). Contrarily, no strong binding was observed for the low-ranking RNAs ($K_{Dapp}$ > 40 μM). Similarly, the $K_{Dapp}$ values of TO-3–N$_3$ also correlated well with the Z-scores of TO-3–N$_3$ of high-ranking RNAs (top 100), as the coefficient was −0.85 (Fig. 5b). These results confirm that our system can provide accurate assessments of different binding modes of ligands and structured RNAs containing G4 structures.

Additionally, we extended this analysis to the commercially available indicators, TO-PRO-1 and TO-PRO3, by measuring their $K_{Dapp}$ values to the 16 selected RNAs and calculating the correlations with the Z-scores of TO–N$_3$ and TO-3–N$_3$, respectively (Figs. 5c, d and S13–S15 and Tables S3 and S4). Regarding TO-PRO-1, the $K_{Dapp}$ values exhibited weak and improved correlations with the Z-scores of TO–N$_3$ ($r = -0.60$) and TO–N$_3$-2 ($r = -0.71$), respectively, indicating that the binding profile of TO–N$_3$-2 may reflect TO-PRO-1 binding by various RNA motifs more accurately (Fig. 5c). Conversely, for TO-PRO-3, there were significant correlations between the $K_{Dapp}$ values and Z-scores of TO-3–N$_3$ ($r = -0.89$) and TO-3–N$_3$-2 ($r = -0.90$) (Fig. 5d). Taken together, these binding profiles will benefit the selection of the proper combinations of target RNA and fluorescent indicators for FID assays.

**Screening of the novel RNA-binding molecules by fluorescent indicator displacement assay using TO-PRO-1 and TO-PRO-3**

Based on the binding profiles of the TO derivatives, we selected the intermediate-affinity-ranked combinations of the indicator and disease-related human pre-miRNAs previously observed to be dysregulated in several tumors, hsa-mir-221, -191, and -21, for the FID assay (Fig. 6)[52–54]. As a high-rank G4 RNA control, hsa-mir-6850 was selected. Additionally, as a low-rank control, the terminal loop motifs from hsa-mir-374a and SARS-CoV-2 RNA (SARS-low) were selected. The predicted RNA secondary structures are shown in Fig. 6a, and the $K_{Dapp}$ values of TO-PRO-1 and TO-PRO-3 to these target and control RNAs are listed. The signal-to-background (S/B) ratios of TO-PRO-1 and TO-PRO-3 for these RNAs are summarized in Fig. 6b. The S/B ratios of the low-rank RNAs were significantly lower than the others. A low S/B ratio is not favorable for performing an accurate FID assay. To identify the small molecules that bind to the target human pre-miRNAs listed above, we employed FID to screen a commercially available chemical library comprising 118 oxidation–reduction compounds (Targetmol) (Supplementary Data 6–8). In this library, chelerythrine chloride (Che)[55–57] is a known intercalating molecule with large π-plane and cationic sites and will be used as a positive control. The fluorescence emission of TOs depends on the RNA binding: free TOs exhibit low fluorescence, although the intensity increases upon RNA binding. Thus, the fluorescence emission of TOs decreases when a test compound interacts with a target RNA via the same site as the fluorescent indicator, thereby identifying it as a hit compound (Fig. 6c). We defined the hit threshold as the mean subtracted by twice standard deviations (mean−2$\sigma$). Through this screen, we identified a total of four hit compounds that disrupted TO–RNA interactions (Figs. 6d, e and S16). Although three of these compounds—baicalein (Bai), myricetin (Myr), and Che—were hits obtained from the assay when using TO-PRO-1, Bai did not meet our selection criteria when TO-PRO-3 was used as the indicator; rather, AS 602801 (AS) became a hit compound (Fig. 6d). This is probably because TO-PRO-3 differs in size and/or fluorescent properties compared with TO-PRO-1, indicating that diverse fluorescent indicators should be included to avoid false negatives and positives. Regarding the hit compounds, Myr[58–60] and Che[55–57] have been reported as DNA or RNA binders, whereas AS has not been reported.

The RNA binding of the four hit compounds was validated by measuring their $K_{Dapp}$ values by fluorescence titrations (Fig. 7). These

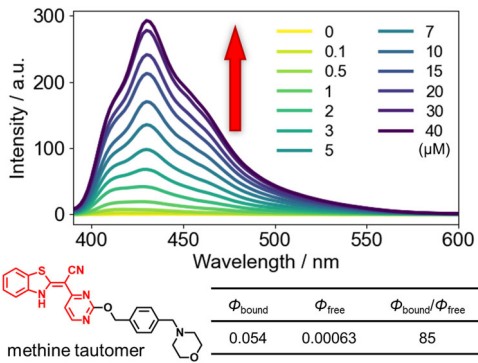

**Fig. 8 | Fluorescence spectra of AS in the titration assay.** Fluorescence spectra of AS (1 μM) in the titration assay of hsa-mir-191 (top, upon addition of 0–40 μM RNA), the methine tautomer structure of AS (bottom left, shown in red), and the quantum yields of AS with and without RNA (bottom right).

experiments revealed that Bai exhibits weak RNA binding ($K_{Dapp} > 40$), indicating that it is a false-positive compound for targeting disease-related human pre-miRNAs when using TO-PRO-1 (Figs. 7a, b and S17). The structurally similar flavonoid, Myr, exhibited moderate binding ($K_{Dapp} = 16–25$) to target RNAs, as the indicators revealed (Figs. 7 and S18). Unexpectedly, Myr bound strongly to hsa-mir-6850, which forms a G4 structure, although it was not identified as a hit compound when TO-PRO-3 was used (Fig. 7a and c). This suggests that Myr and TO-PRO-3 might have different binding sites. When using low-rank RNAs, Myr exhibited weak RNA binding ($K_{Dapp} > 40$) even though the indicators exhibited positive. Moreover, we observed that Che was bound to all the RNAs ($K_{Dapp} = 2.6–16$) though the indicators exhibited negative for low-rank RNAs (Figs. 7 and S19). Overall, predictably unreliable results were obtained when low-rank RNAs were used. The precisions of demonstrating the reliability of the assay data across the investigated RNAs became worse as the RNA ranking decreased (Fig. S20), suggesting that our binding profiles offered insight into the selection of applicable RNA targets for indicators in FID assays.

Finally, we observed AS binding to hsa-mir-191, -21, and -6850 ($K_{Dapp} = 14, 20,$ and $4.5,$ respectively). Interestingly, this compound exhibited strong light-up properties (Figs. 8 and S21): although free AS exhibited almost no fluorescence ($\Phi_{free} = 0.00063$), strong fluorescence was observed after RNA binding ($\Phi_{bound} = 0.054$). The methine tautomer[61] likely contributes to this light-up property. TO-PRO-1 could not detect the RNA binding of this compound because of the interference of its strong light-up property at a similar wavelength range with the detection of the fluorescence originating from TO-PRO-1. These characteristics make AS an interesting seed compound for developing novel RNA binders and fluorescence probes.

## Conclusions

We developed the large-scale analytical platform for investigating small molecule–RNA interactions by subjecting the small molecules to FOREST. The affinity profiles generated by FOREST include not only high-affinity interactions but intermediate and low-affinity ones on the wide range of RNA structures that were derived from naturally occurring sequences. Additionally, compared with methods using massively parallel DNA sequencing, FOREST—by using microarray analysis to determine the binding affinities of RNA structure libraries—presents the affinity profiles of small molecules without any structure-dependent amplification bias[30]. First, we validated our system using the unpaired G-specific binding property of the G-clamp (Figs. 2 and 3). The FOREST system ranked the G-clamp bindings of high-, intermediate-, and low-affinity RNA targets. The mutation experiments using rank 1 RNA (hsa-mir-4520-1 loop) showed that G-clamp forms hydrogen bonds with specific Gs. For further studies that will reveal detailed complex structures, such as X-ray crystallography or

NMR, the large-scale affinity profile would help select suitable sequences for structure determination because the difficulty of these structural analyses differs depending on the sequence. Second, we generated the binding profiles of the TO derivatives using this platform (Figs. 4 and 5). Employing FOREST profiling, G4 structures, which are susceptible to bias by sequencing-based methods, were evaluated and ranked as top-tier interactors of the TO derivatives. Additionally, the analysis of the affinity profiles reveals a binding preference of the TO derivatives for RNA motifs containing G-rich terminal loops, internal loop G/A bases, and/or G/C-rich stem structures (Fig. 4c–e).

The library-wide binding landscape and profiles were also applicable to commercially available fluorescent indicators, TO-PRO-1 and TO-PRO-3, for FID assay (Fig. 6). Since our knowledge of fluorescent indicator–RNA combinations remains limited, the profiles generated by this system can benefit the selection of optimal combinations and further expand the repertoire of target RNA sequences for FID assays. In this study, we conducted FID assays using different ranks of RNA and TO-PRO-1 or TO-PRO-3 as target RNA and fluorescent indicators, respectively. The FID assays using these indicators and low-rank RNAs could not provide accurate hit compounds, while high- and intermediate-rank RNA provided reliable results (Fig. 7), demonstrating that our binding profiles are valuable for selecting applicable combinations for the FID assay. Moreover, we demonstrated the utility of this screening approach by identifying AS 602801 as an RNA binder that binds hsa-mir-191, -21, and -6850 with remarkable light-up properties (Figs. 7a, 8, and S21). Considering AS 602801 was identified using only TO-PRO-3 and FID assays have the limitation that they can basically detect compounds with similar binding sites and modes, the use of multiple fluorescent indicators is recommended for FID assays. In addition, the development of new fluorescent indicators that differ from known ones will be important to address the limitation in hit compound types. For example, indicators that conjugate fluorescent units whose fluorescence changes with an RNA binding event to an RNA-binding molecule[45,62,63] with various binding modes are expected to provide new hit compounds that have been overlooked by existing indicators in FID assays. In this case, FOREST will be valuable for obtaining RNA-binding information, designing the conjugated indicator, and understanding its binding preference.

The FOREST system in this study provides the basis for future efforts to identify new small molecule–RNA interactions, investigate the binding profiles and selectivities of various RNA-binding molecules, and aid the design of novel RNA-binding molecules through FID assays.

## Methods

### In silico RNA motif extraction

All motifs, including human pre-miRNA in library-1 and -2 were extracted from miRBase as detailed previously[30]. To design library-2, the human pre-miRNA motifs were filtered based on length (<107 nt), with 1804 species collected in total. Next, we obtained RNA secondary structure datasets as determined by SHAPE-MaP or DMS-MaPseq with structural analysis[64,65]. Predicted structures and conserved elements of SARS-CoV2 were obtained from a published study[66]. From the collected datasets, we divided long continuous RNAs into terminal motifs and defined them as structural units using FOREST.py (https://github.com/KRK13/FOREST2020). In total, 1099 motifs were collected from the transcripts of SARS-CoV2 and Influenza A viruses. As controls, selected RNA structural motifs, aptamers, and defective mutants were collected and loaded into the libraries.

### Design of a template pool of RNA structure library and DNA barcode microarray

Multiplexed single-stranded DNA sequences were used as templates for RNA probes in the library. The extracted RNA motifs were attached with T7 promoter, RNA barcodes, and stabilizing stem sequences for detection and hybridization to the DNA barcode microarray as previously described[30]. The ssDNA templates were synthesized by SurePrint oligonucleotide library

synthesis (Agilent Technologies). The size of the oligo template was limited to 170 nt for RNA structure library-1 and 190 nt for library-2. After assigning barcodes to RNA structures, the DNA reverse complementary strands of RNA barcodes were used by SureDesign (Agilent Technologies), a custom CGH array design service, to synthesize DNA barcode microarrays. The probe replication factor was set to 5× and 3×.

### 3'-Terminal labeling with Cy5 or Cy3
All RNA probes in the RNA structure libraries were labeled with a fluorescent dye at the 3' end. Ten micromolar RNA structure library, 100 μM pCp-Cy5 or pCp-Cy3 (Jena Bioscience), and 0.5 U/μL T4 RNA Ligase (Thermo Fisher Scientific) were mixed in 100 μL of 1× T4 Ligase Buffer (Thermo Fisher Scientific). The mixture was incubated at 16 °C for 48 h on a ThermoMixer (Eppendorf) with ThermoTop (Eppendorf). After incubation, the labeled RNA was purified using Zymo RNA Clean and Concentrator (Zymo Research) and stored at −28 °C until use.

### Synthesis of $N_3$-modified RNA binders
The $N_3$-modified G-clamp–$N_3$, TO–$N_3$, and TO-3–$N_3$ were synthesized using $N_3$–$PEG_3$–$NH_2$ as an $N_3$ linker after preparing the carboxylic acid intermediates (Supplementary Methods and Schemes S1–S3). TO–$N_3$-2 and TO-3–$N_3$-2 were synthesized using $N_3$–$PEG_4$–NHS ester as an $N_3$ linker after preparing the amine intermediates (Supplementary Methods and Schemes S4 and S5)[67,68].

### RNA pull-down
The RNA structure library was prepared in 1× Binding buffer (20 mM phosphate pH 7.0, 20 mM NaCl, 80 mM KCl)[30]. For folding, RNA was heated at 95 °C and cooled to 4 °C on a ProFlex Thermal Cycler (Thermo Fisher Scientific) with a ramp rate of −6 °C/s. During the folding step, 100 pmol of small molecules and 50 μL of Streptavidin Mag Sepharose (Cytiva) were mixed in 900 μL of 1× Binding buffer to prepare the small molecule-conjugated beads. The mixture was incubated on a ThermoMixer (Eppendorf) at 25 °C for 60 min with vortex mixing at 1200 rpm. The tube was placed on a magnetic rack to remove the supernatant and 1 μg of the refolded RNA structure library in 1 mL of 1× Binding buffer was added. A mixture containing only the beads was prepared as a control for background subtraction. The mixture was incubated on a ThermoMixer at 25 °C for 60 min with vortex mixing at 1200 rpm. The mixture was washed three times with 1× Binding buffer when the reaction ended. Two hundred microlitres of 1× Elution buffer (1% SDS, 10 mM Tris–HCl, 2 mM EDTA) was added to the magnetic beads, and the mixture was heated at 95 °C for 3 min. The bound RNA structures were collected from the supernatant by removing the magnetic beads and purified with phenol-chloroform extraction and ethanol precipitation.

### Hybridization and microarray scanning
Eighteen microlitres of the bound RNA structures were mixed with 4.5 μL of 10× Blocking Agent (Agilent Technologies) and 22.5 μL of Hi-RPM Hybridization Buffer (Agilent Technologies). The samples were incubated for 5 min in a heat block set at 104 °C, then rapidly cooled and incubated for 5 min in ice water. The samples were applied to an 8 × 60K Agilent microarray gasket slide (Agilent Technologies). The prepared gasket slide and CGH custom array 8 × 60K (Agilent Technologies) were assembled with SureHyb. Hybridization was performed for 20 h at a temperature of 55.5 °C at 20 rpm. The microarray slide was washed for 5 min with Gene Expression Wash Buffer 1 (Agilent Technologies) in a glass container at room temperature following hybridization. The microarray slide was moved to a glass container containing Gene Expression Wash Buffer 2 (Agilent Technologies), which was immersed in a thermostatic bath at 37 °C. The washing step was performed for 5 min. Fluorescence scanning was performed on the microarray, and fluorescence image data were acquired using SureScan (Agilent Technologies). The acquired images were converted to numeric fluorescence intensities for each spot by Feature Extraction (Agilent Technologies) and GeneSpringGX (Agilent Technologies).

### Calculation of binding intensity
The binding intensities of each RNA structure were calculated by subtracting the fluorescence intensities of the no-ligand control samples. To alleviate the effect of undesired interactions with the RNA barcode, we calculated the mean fluorescence intensities of each RNA structure from the intensities of three RNA probes that had the same RNA structure but different RNA barcodes. For this reason, we filtered the maximum and minimum values from a set of five intensities.

### Statistics
For testing statistical significance, the two-tailed Brunner–Munzel test with Bonferroni correction was performed using Julia 1.6. standard error (SE) was calculated using the three probes of the RNA structure library. The binding strength is normalized as a Z-score using Eq. (1): $\mu$ is the mean value of the library population, $\sigma$ is the standard deviation, and $x$ is the binding intensity of each probe in the library.

$$Z\,score_x = \frac{x - \mu}{\sigma} \quad (1)$$

### Fluorescence binding assay
A solution (100 μL) of the binder (0.01 or 0.1 μM for G-clamp, 0.1 μM for TO–$N_3$ and TO-PRO-1, 1 μM for TO-3–$N_3$, 0.1 or 0.5 μM for TO-PRO-3) in 1x phosphate buffer (1% DMSO, 20 mM phosphate, 20 mM NaCl and 80 mM KCl) was transferred to a micro quartz cell with a 1-cm path length. Serial aliquots of a concentrated solution of RNA in 1× buffer were added to the binder solution and allowed to equilibrate for 2 min. The excitation wavelength was set at 360 nm for G-clamp, 501 nm for TO–$N_3$ and TO-PRO-1, 623 nm for TO-3–$N_3$ and TO-PRO-3, and the emission was recorded at 20 °C. Fluorescence measurements were performed with a JASCO-6500 spectrofluorometer (JASCO, Tokyo, Japan).

The data from the titrations were analyzed according to the independent-site model by non-linear fitting to Eqs. (2) or (3), in which $F_0$ is the initial fluorescence intensity in the absence of RNA, $Q$ ($=F_{max}/F_0$) is the fluorescence enhancement upon saturation, $A = K_{Dapp}/C_{ligand}$ and $X = nC_{RNA}/C_{ligand}$ ($n$ is the putative number of binding sites on RNA and $n = 1$ was used)[69]. The parameters $Q$ and $X$ were determined by Kaleida-Graph (Synergy Software, PA). The $K_{Dapp}$ values in the main text show the mean values of two or three experiments.

$$F/F_0 = 1 + (Q - 1)/2\{A + 1 + X - [(X + 1 + A)^2 - 4X]^{1/2}\} \quad (2)$$

$$or \; \Delta F = F - F_0 = F_0(Q - 1)/2\{A + 1 + X - [(X + 1 + A)^2 - 4X]^{1/2}\} \quad (3)$$

### SPR analysis
Immobilization: 5′-biotinylated RNA (hsa-mir-4520-1 loop, mutG2A, or mutG7A) was diluted to 1 μM in 1× Binding buffer (20 mM phosphate pH 7.0, 20 mM NaCl, and 80 mM KCl), and the solution was heated at 95 °C for 5 min and cooled on ice. The folded RNAs were injected over a streptavidin-coated sensor chip (Series S Sensor chip SA, Cytiva) at 60 μL/min to reach an immobilized level of 1481, 1379, and 1387 RU for the hsa-mir-4520-1 loop, mutG2A, and mutG7A, respectively.

Binding analysis by single-cycle kinetics: the RNA binder (G-clamp-$N_3$) in 1× Binding buffer (20 mM phosphate pH 7.0, 20 mM NaCl, and 80 mM KCl) was injected at increasing concentrations (100, 200, 300, 400, and 500 nM for hsa-mir-4520-1 loop, 20, 40, 60, 80, and 100 nM for mutG2A, or 1, 2, 3, 4, and 5 μM for mutG7A) to the RNA-immobilized sensor surface without a regeneration step between each concentration. The RNA binder was injected with a flow rate of 60 μL/min, contact time of 30 s, and dissociation time of 120 s using the running buffer at 25 °C. All sensorgrams

were corrected by subtracting the blank flow cell and buffer injection responses. All kinetics were obtained by Biacore T200 evaluation software.

Binding analysis by multi-cycle kinetics: the RNA binder (G-clamp-N$_3$) in 1× Binding buffer (20 mM phosphate pH 7.0, 20 mM NaCl, and 80 mM KCl, 1%DMSO) was injected at increasing concentrations (1, 2, 3, 5, 10, 20, 30, and 50 µM for mutG7A) to the RNA-immobilized sensor surface with a regeneration step between each concentration. The RNA binder was injected with a flow rate of 60 µL/min, contact time of 30 s, and dissociation time of 120 s using the running buffer at 25 °C. A regeneration step was conducted with a flow rate of 60 µL/min and contact time of 30 s using 1 M NaCl solution. All sensorgrams were corrected by subtracting the blank flow cell and buffer injection responses. SPR response values at 20 min were used to compute the $K_{Dapp}$ value using the 1:1 binding equation $\{y = (B_{max} + x)/(K_{Dapp} + x)\}$, where $y$ is the SPR response, $B_{max}$ is the maximum SPR response, $K_{Dapp}$ is the apparent dissociation constant, and $x$ is the concentration of the added RNA binder.

### RNA secondary structure prediction and visualization

The forna website[70] was used to generate illustrations of the RNA secondary structures predicted by RNAfold 2.4.13 in the ViennaRNA package[49] with the temperature set to 25 °C. The RNA structures extracted from the long transcripts (5' UTR and HIV-1 genome) included in library-2 were taken from a previous study[30].

### Structural preference analysis

Following previous studies[71], secondary structure prediction of RNA motifs in the RNA structure library was performed by RNAsubopt 2.4.13 in the ViennaRNA package[49] with parameters set to the following: (command: RNAsubopt --temp=25 --stochBT=30). Each nucleotide (A, G, U, C) of each base pair state (ssRNA or dsRNA) or each structural motif (terminal loop, inner loop, or stem) was counted using the secondary structures generated by RNAsubopt as input.

### Molecular modeling

The RNA 3D structures were predicted using RNAComposer[50,51]. The energy minimization of complex structures between RNA and G-clamp–N$_3$ was performed using MacroModel (Schrödinger) after setting G-clamp–N$_3$ to face the G base so that hydrogen bonds could be formed. OPLS3e and water were used as the force field and solvent, respectively.

### FID assay

Fluorescence intensities in FID assays were measured with a microplate reader Infinite® 200 PRO (TECAN Group Ltd., Mannedorf, Switzerland) using i-control® and LBS-coated Optiplate™-96F as 96-well plates. Buffer solution (20 mM phosphate pH 7.0, 20 mM NaCl, 80 mM KCl) was added to each well (49.5 µL for blank well and negative control well, 49 µL for positive control well and sample well), followed by the addition of 0.25 µL of ligand solution (20 µM for TO-PRO-1 and 100 µM for TO-PRO-3) to each well except for blank wells. RNA solution (0.5 µL of 10 µM for TO-PRO-1 and 50 µM for TO-PRO-3) in Binding buffer was dispensed in positive control and sample wells. DMSO was added to the control (negative and positive, 0.25 µL) and blank (0.5 µL) wells; while 0.25 µL of compound solution in DMSO (1 mM, Targetmol) was added to each sample well and mixed with RNA-ligand solutions. Fluorescence intensities of the mixtures were measured after incubating for 30 min. The excitation wavelength was set at 485 nm for TO-PRO-1 or 620 nm for TO-PRO-3. Normalized fluorescence intensity ($F$) was calculated using Eq. (4) described below:

$$Normalized\ F = \frac{F_{(indicator+RNA+test\ compounds)} - F_{(buffer+indicator)}}{F_{(indicator+RNA)} - F_{(buffer+indicator)}} \quad (4)$$

Hits were selected based on a reduction of TO-PRO-1 or TO-PRO-3 signal by less than two standard deviations ($2\sigma$) from the mean. Normalized fluorescence intensities >1.5 were excluded from calculations for the mean and σ.

### Calculation of fluorescent quantum yield

The fluorescent quantum yields (QY) of AS 602801 in the presence of RNA were calculated using quinine sulfate in 0.1 M H$_2$SO$_4$ as a standard ($\Phi = 0.55$). Absorbance and fluorescence values were recorded 3 min after mixing RNA and AS 602801. For calculating QY, conditions for absorbance measurement were as follows: [AS 602801] = 2.5 µM, [RNA] = 5 µM, and $\varepsilon 366$; and for fluorescence measurement: [AS 602801] = 1 µM, [RNA] = 2 µM, emission spectrum area of 380–600 nm was used for integration. QY values were calculated according to Eq. (5):

$$\phi_{sam.} = \phi_{ref.} \times \frac{\varepsilon_{ref.}}{\varepsilon_{sam.}} \times \frac{c_{ref.}}{c_{sam.}} \times \frac{(n_{sam.})^2}{(n_{ref.})^2} \times \frac{F_{sam.}}{F_{ref.}} \quad (5)$$

where $\Phi_{sam.}$ is quantum yield of the sample, $\Phi_{ref.}$ is the quantum yield of the reference compound, $\varepsilon_{sam.}$ is the molar extinction coefficient of the sample, $\varepsilon_{ref.}$ is the molar extinction coefficient of the reference compound, $c_{ref.}$ is the concentration of the reference compound, $c_{sam.}$ is the concentration of the sample, $n_{sam.}$ is the refractive index of the sample solution, $n_{erf.}$ is the refractive index of the reference solution, $F_{sam.}$ is the fluorescence intensity of the sample solution, and $F_{ref.}$ is the fluorescence intensity of the reference solution.

### Reporting summary

Further information on research design is available in the Nature Portfolio Reporting Summary linked to this article.

### Data availability

The datasets of FOREST are available in Supplementary Data 1–5. The compound structures from the chemical library are available in Supplementary Data 6. The datasets of the FID assay are available in Supplementary Data 7 and 8.

### Code availability

The custom codes for terminal motif extraction and designing the RNA structure library are available on the Github page (https://github.com/KRK13/FOREST2020/). The other codes used in this study are available from the corresponding authors upon request.

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

## Acknowledgements
We thank Kelvin Hui (Kyoto University) for critical reading of the manuscript. We thank Yoshikazu Tanaka, Kanami Takahashi and Eiko Hanzawa (Tohoku University) for supporting SPR analysis and Asako Murata (Kyushu University) for advice regarding SPR analysis. This work was supported in part by Grant-in-Aid for Scientific Research on Innovative Areas "Middle Molecular Strategy" (No. JP15H05838 to F.N.), "ncRNA neotaxonomy" (No. JP17H05601 to H.S.), "Frontier Research on Chemical Communications" (No. JP20H04762 to K.O.), Transformative Research Areas (A) "Biophysical Chemistry for Material Symbiosis" (No. JP23H04051 to K.O.), Scientific Research (B) (JP19H02845 to K.O. and JP20H02855 and JP23H02076 to F.N.), Specially Promoted Research (No. JP20H05626 to H.S.), Challenging Exploratory Research (No. JP19K22387 to H.S. and No. JP21K19038 to F.N.) from the Japan Society for the Promotion of Science (JSPS); Japan Science and Technology Agency (JST) FOREST program (No. JPMJFR2002 to K.O.) and SPRING program (No. JPMJSP2114 to R.N.); the Takeda Science Foundation (K.O.), the Uehara Memorial Foundation (K.O.), the Noguchi Foundation (K.O.), the Tokyo Biochemical Research Foundation (K.O.), the Naito Foundation (K.R.K.), the Mitsubishi Foundation (H.S.), and the research program of "Crossover Alliance to Create the Future with People, Intelligence and Materials" from MEXT, Japan.

## Author contributions
K.O. and K.R.K. designed the experiments. K.O., H.S. and F.N. mentored the research. R.N., H.M., K. Ojima, and S.I. synthesized compounds. R.N., K.R.K., E.M. and M.O. performed analytical experiments. R.N., K.O., K.R.K., and E.M. analyzed the results. R.N., K.O. and K.R.K. mainly wrote the manuscript. All authors discussed the results and provided feedback on the study and manuscript.

## Competing interests
K.R.K. and H.S. own shares of xFOREST Therapeutics Co., Ltd. All other authors declare no competing interests.
