## [Peer Review File · Communications Chemistry]

Reviewers' comments:

Reviewer #1 (Remarks to the Author):

The work reported by Nagasawa and co-workers describes the application of a previously published protocol called FOREST to the study of potential fluorophores to be used in FID assays and to the screening of a library of compounds as RNA binders. In this context, they identified first two intercalators able to bind various RNA sequences/structures that have been ranked based on the affinity. These intercalators have then be employed in a FID assay to identify binders of the best targets.

The work is well conducted and described, however this reviewer has major concerns regarding the advantages and the conclusions of this work. The authors state “Additionally, compared with methods using massively parallel DNA sequencing, FOREST—by using 244 microarray analysis to determine the binding affinities of RNA structure libraries—presents the affinity profiles of small 245 molecules without any structure-dependent amplification bias”. This concern the previous works by the same authors and it is correct, but here there is a major bias coming from the screened compounds. Indeed, FID assays better identify intercalator compounds and induce a bias in the compounds that are identified. Even if the authors find that more than one intercalator need to be used for the accuracy this is not enough for the identification of RNA binders that would not interact by intercalation and that would be more interesting because more likely to be selective.

The second bias is the chosen library. The list of compounds is the one on Figure S14, this reviewer could not fin any detail about the choice of the library, the compounds (the name of the compounds is hard to read because of the resolution of the figure) and if there were more than the ones indicated. An explanation of the chosen library and of the intercalating potential of the compounds should be provided to appreciate the reliability of the screening and of the results.

Finally, it is hard to appreciate the advantage of such an assay compared to other very well known FID assays that can also be performed in high-throughput. The main advantage would be to screen a library of compounds against a library of RNA structures, however here only intercalators can be identified and this is indeed what happens when looking at the structure of the identified hits.

In conclusion, despite a thorough work and a large number of experiments, this reviewer recommends publication in a more specialized chemical biology journal.

Reviewer #2 (Remarks to the Author):

This manuscript describes the large-scale screening or RNA binding to specific molecules using the FOREST techniques and the pull-down of RNAs from RNA libraries with biotin-labeled molecules. I reviewed this manuscript (or a related one) in another journal.

I raised the following points for the revision to make this manuscript easier for most readers:

The authors described on page five that “Five or more unpaired Gs were mainly observed in high-ranking RNAs (1–180), and the percentage decreased gradually as the rank decreased. Contrarily, few RNAs without any or only a single unpaired Gs were observed in the high-ranking group, and the percentage gradually increased as the rank decreased. These results corresponded to the fact that G-clamp mostly

recognizes G base in the ssRNA regions”.

According to the chemical structure of the G-clamp, this molecule can bind to the G through H-bonding as the molecule has similar H-bonding groups as cytosine. However, the G-clamp can bind by H-bond with only one guanine. Even if we assume that the G-clamp could be stacked between two guanines, the maximum number of guanines necessary for the G-clamp binding would be three. Why were five or more unpaired Gs mainly observed in high-ranking RNAs?

Fluorescence titration seems to provide over-estimated K_d , especially when the binding of RNA quenched the small-molecule fluorescence. This is probably due to the presence of multiple pathways of fluorescence quenching. I encourage authors to determine the K_d by other methods if the discussion of K_d was critical for these analyses.

Docking studies are helpful in formulating ideas, but more is needed to explain how the bound structure is formed. Are there any other possible bound structures? How are stable structures produced with other mutated RNA sequences?

I still need to understand why five or more Gs are ranked in the top selected RNAs. If the G-clamp can bind to the RNA by H-bonding, authors may be able to see the specificity with a single G in the ssRNA region.

Reviewer #3 (Remarks to the Author):

The authors have developed a new system for large scale analysis of RNA-Small Molecule interactions to provide information that can aid discovery of novel RNA-binding molecules. This is an interesting approach that could prove to be timely and useful in the development of small molecules that can interact with and modulate structured RNAs. I think the manuscript well-crafted and should be good to publish after minor revisions.

Abstract: Looks good. Provides a good introductory statement regarding RNA and small molecule interactions, an overview of what was done in the study, and what the results and their implications are.

Introduction: This section reads well and includes all necessary components. Good background of RNA small molecules work including work by Disney and Sugimoto. Nice introduction to their FOREST method that was the spring board to the approach used in this study. Finishes with a nice synopsis of what their study has done and how it is useful to the community.

- Line 33. “small molecule-binding” is more appropriate than “RNA-binding”

Results and Discussion:

- Line 89 through 94 should be in the methods.
- When discussing the analysis of g-clamp interactions starting at line 96, it would be nice to have a brief outline of RNA ranking in the main text (something simple like what is in the figure caption). It would also be great to know which RNA folding algorithm was used or where the structures came from.

- Lines 114-115, why are the only RNA motifs with a common stem used for Kd calculations.
- Line 146, why was Library 1 not used for TO and TO-3? Are the pre-miRNA loops in the Library 2 to the same as those used in Library 1?
- Line 175 and 183, why were viral RNAs only analyzed with TO-PRO derivatives?
- Lines 194-213. The structure of this section is a bit confusing. The figure panels are not mentioned in order so maybe reorganize the figure or reorganize the main text so there is better flow. If the main text is left as is, maybe have the structures as A, the S/B plots as B, the FID assay as C, and the compounds and tables as D or D-G in the figure.
- Lines 226-238 are a bit confusing to me. This section kind of feels out of place compared to the rest of the paper. The AS spectrum fits better than the Che spectra as it relates more to the potential development of new probes for these kind of FID experiments.

Conclusion:

Solid conclusion section that reiterates the main points of the paper and the potential applications of the FOREST system.

Methods:

For the structural preference analysis, why was RNAsubopt used instead of the normal MFE fold from RNAfold?

For FID assay calculations, why was a reduction of less than a standard deviation used and normalized values greater than 1.5 excluded?

Figures

Figure 1: Good flow to the figure. Pretty understandable without the caption, however, a bit of additional information would be nice in the caption.

Figure 4: I am thinking the definition of hairpin here is referring to the terminal loop. Clarification of this would be great.

Figure 5: I would suggest breaking this figure up and give each plot a different letter so a direct reference to each plot can be made in the text. I would also keep all the colors the same between A and B.

Figure 6: The caption does not explain this figure very well. Similar to figure 5, maybe the tables need individual letters so they can be referenced directly. The caption needs to be more descriptive as to what the numbers in the table actually represent. In panel b, there is a missing hyphen in TO-3-N3-2

Point-by-point responses to the reviewers' comments

Reviewer 1:

We express our gratitude to Reviewer 1 for his/her important comments and suggestions to improve the quality of our manuscript. We reconsidered the advantage of our system and potential bias of FID assay and revised our manuscript with additional sentences. We hope that these revisions will be to his/her satisfaction.

1-1: The work is well conducted and described, however this reviewer has major concerns regarding the advantages and the conclusions of this work. The authors state “Additionally, compared with methods using massively parallel DNA sequencing, FOREST—by using microarray analysis to determine the binding affinities of RNA structure libraries—presents the affinity profiles of small molecules without any structure-dependent amplification bias”. This concern the previous works by the same authors and it is correct, but here there is a major bias coming from the screened compounds. Indeed, FID assays better identify intercalator compounds and induce a bias in the compounds that are identified. Even if the authors find that more than one intercalator need to be used for the accuracy this is not enough for the identification of RNA binders that would not interact by intercalation and that would be more interesting because more likely to be selective.

[Answer-1-1] Thank you for pointing out the bias of the FID assay. We agree with the reviewer's opinion that intercalating fluorescent indicators basically detect intercalators as hit compounds, and the data in our manuscript are insufficient for the full identification of RNA binders. Accordingly, we have revised our manuscript to describe the limitations of the current FID assay system and future perspectives of FOREST in the discussion as follows.

In this study, we mainly report large-scale analysis of small molecule-RNA interactions. We introduced the structure-dependent amplification biases in profiling the small molecule binding by massively parallel DNA sequencing. The FOREST provides the affinity profiles of small molecules without any structure-dependent amplification bias. Reviewer 1 has agreed with this point. Based on the obtained information, we have discussed the binding preference of RNA-binding molecules and used the information for an FID assay as one of the possible applications. We selected TO-PRO-1 and TO-PRO-3 as indicators because they are commercially available and often used for FID assay. We considered that the large-scale data of the binding property and selectivity for these indicators would be useful, and this approach would be a good example for our system.

As reviewer 1 stated, the FID assays using TO-PRO-1 and TO-PRO-3 are insufficient for identifying RNA binders that would not interact by intercalation. To overcome this bias, we are currently attempting to develop new fluorescent indicators with RNA-binding information obtained by FOREST. We designed a new indicator conjugated thiazole orange (TO) derivative and an RNA binder

(Supplemental Review Figure 1). For example, when G-clamp was conjugated with TO derivatives, the TO-G-clamp product had a light-up property and the binding selectivity across various RNAs was similar to that of G-clamp. Research on a similar concept has also been reported by Shibata et al. (ref45, Chem. Commun. 2023). The details of the development of new fluorescent indicators are beyond the scope of this study; therefore, they will be reported in our next manuscript.

Accordingly, we have added the following sentences that describe the limitations of FID assay and future possibilities of the FOREST system in the revised manuscript.

Conclusion, 2nd paragraph

“Considering AS 602801 was identified using only TO-PRO-3 and FID assays have the limitation that they can basically detect compounds with similar binding sites and modes, the use of multiple fluorescent indicators is recommended for FID assays. In addition, the development of new fluorescent indicators that differ from known ones will be important to address the limitation in hit compound types. For example, indicators that conjugate fluorescent units whose fluorescence changes with an RNA binding event to an RNA-binding molecule^{45,66,67} with various binding modes are expected to provide new hit compounds that have been overlooked by existing indicators in FID assays. In this case, FOREST will be valuable for obtaining RNA-binding information, designing the conjugated indicator, and understanding its binding preference.”

A new fluorescent indicator (future plan)

TO derivatives

Supplemental Review Figure 1. Schematic of a new fluorescent indicator with RNA-binding information.

1-2: The second bias is the chosen library. The list of compounds is the one on Figure S14, this reviewer could not find any detail about the choice of the library, the compounds (the name of the compounds is hard to read because of the resolution of the figure) and if there were more than the ones indicated. An explanation of the chosen library and of the intercalating potential of the compounds should be provided to appreciate the reliability of the screening and of the results.

Thank you for pointing out the lack of information regarding the compound library. As suggested, library information is crucial for evaluating the reliability of the screening results. We have added the figure of the compound structures in the library (Supplementary figure of the chemical library) and an

Excel file of FID assay results (Supplementary list of FID). We used a simple and commercially available library as a model FID assay. We think chelerythrine chloride (Che) is the most likely intercalator molecule with a large π plane and cationic site and will be a positive control. Accordingly, we have added the following sentences.

Result - Screening of the novel RNA-binding molecules by fluorescent indicator displacement assay using TO-PRO-1 and TO-PRO-3, 1st paragraph

“To identify the small molecules that bind to the target human pre-miRNAs listed above, we employed FID to screen a **commercially available** chemical library comprising 118 oxidation–reduction compounds (Targetmol) (**Supplementary figure of chemical library and supplementary list of FID**). **In this library, chelerythrine chloride (Che) is a known intercalating molecule⁵⁵⁻⁵⁷ with large π -plane and cationic sites and will be used as a positive control.**”

1-3: Finally, it is hard to appreciate the advantage of such an assay compared to other very well known FID assays that can also be performed in high-throughput. The main advantage would be to screen a library of compounds against a library of RNA structures, however here only intercalators can be identified and this is indeed what happens when looking at the structure of the identified hits.

[Answer-1-3] Thank you for your valuable comment. The FOREST system is advantageous for analyzing large-scale datasets of diverse RNA structures derived from naturally occurring sequences. In this study, we investigated the RNA-binding profiles of intercalating fluorescent indicators to demonstrate the effectiveness of the FOREST system. As shown in the G-clamp example, this system can also be used with hydrogen bond-forming binding molecules. As we explained in Answer-1-1, we can develop the nonintercalating fluorescent indicators with RNA-binding information by conjugating the fluorescent unit. Therefore, obtaining the RNA-binding profiles of new indicators by FOREST will be useful information to understand the binding preference and select applicable combinations for the FID assay.

In this study, we discussed the binding preference of RNA-binding molecules based on the data of large-scale analysis and performed an FID assay using the obtained RNA-binding profiling to demonstrate the effectiveness of the FOREST system. Analysis of the complex structures by X-ray crystallography or NMR is also important to understand the accurate complex structure and develop RNA-binding molecules. We believe that the obtained RNA-binding profiling by FOREST would help select suitable sequences for analysis because the difficulty of these structural analyses differs depending on the sequence (i.e., it allows for the selection of the most suitable sequences for structural analysis from among the various RNA sequences that bind to target molecules). Therefore, we have added the following sentences.

Conclusion, 1st paragraph

“For further studies that will reveal detailed complex structures, such as X-ray crystallography or NMR, the large-scale affinity profile would help select suitable sequences for structure determination because the difficulty of these structural analyses differs depending on the sequence.”

.....

Reviewer 2:

We would like to thank reviewer 2 for the useful comments and suggestions to improve the quality of our manuscript. As suggested, we reconsidered the explanation of G-clamp selectivity, did additional experiments for the binding affinity of G-clamp, and did the molecular modelling according to the comments. We hope that these revisions will be to his/her satisfaction.

2-1: The authors described on page five that “Five or more unpaired Gs were mainly observed in high-ranking RNAs (1–180), and the percentage decreased gradually as the rank decreased. Contrarily, few RNAs without any or only a single unpaired Gs were observed in the high-ranking group, and the percentage gradually increased as the rank decreased. These results corresponded to the fact that G-clamp mostly recognizes G base in the ssRNA regions”.

According to the chemical structure of the G-clamp, this molecule can bind to the G through H-bonding as the molecule has similar H-bonding groups as cytosine. However, the G-clamp can bind by H-bond with only one guanine. Even if we assume that the G-clamp could be stacked between two guanines, the maximum number of guanines necessary for the G-clamp binding would be three. Why were five or more unpaired Gs mainly observed in high-ranking RNAs?

[Answer-2-1] Thank you for pointing out the lack of explanation. We think that G-clamp does not recognize all Gs on the loop (G-clamp recognizes specific Gs) from the mutation experiments in Figure 3C. The high number of G bases in the ssRNA region of high-ranking RNAs probably increased the probability of the presence of G bases that strongly bind to G-clamp. In addition, we think G-clamp binds not only specific unpaired G by hydrogen bonds but also G4 or G4-like structure by intercalation. Tricyclic compounds with cationic parts can bind to G4. For example, acridine derivatives, such as BRACO-19 (PNAS, 2001, 98, 4844–4849), are well known. G-clamp is also a tricyclic compound with a cationic part and thus has the potential for G4 binding. We showed that the K_{Dapp} values of the hsa-mir-6850 loop (rank 28, known G4 RNA) and G4_(GGGU)6 (rank 38, G4 control) were 0.19 and 0.15 μ M, respectively. From kernel density estimation of G-clamp Z-scores, G4 control was significantly enriched, as shown in Supplemental Review Figure 2.

From these two reasons, (1) increased probability of the existence of G bases that strongly bind to G clamps and (2) G4 or G4-like structure binding, and G4 or G4-like structures are often formed when multiple Gs exist in a single-stranded region, we think that five or more unpaired Gs were mainly observed in high-ranking RNAs (1–180).

We partly discussed it in Large-scale analysis of the interaction of G-clamp-N₃ with Library-1, 2nd paragraph.

“The high number of G bases in the ssRNA region of high-ranking RNAs probably increased the probability of the presence of G bases that bind to G-clamp strongly.”

We have added the following sentences for further explanation.

Result - Large-scale interaction analysis of G-clamp-N₃ with library-1, 2nd paragraph

“The K_{Dapp} values of the hsa-mir-6850 loop (rank 28) and G4_(GGGU)₆ (rank 38) were 0.19 and 0.15 μ M, respectively. This may be because G-clamp intercalated on G4 RNAs. RNAs with five or more unpaired Gs are particularly enriched in Figure S3 because G-clamp binds not only specific unpaired Gs but also G4 or G4-like structures, and such structures are often formed when multiple Gs exist in a single-stranded region.”

Supplemental Review Figure 2. Kernel density estimation of G-clamp Z-scores. The p -value were determined by two-tailed Brunner–Munzel test.

2-2: Fluorescence titration seems to provide over-estimated Kd, especially when the binding of RNA quenched the small-molecule fluorescence. This is probably due to the presence of multiple pathways of fluorescence quenching. I encourage authors to determine the Kd by other methods if the discussion of Kd was critical for these analyses.

[Answer-2-2] Thank you for your comment. As suggested, we have examined K_{Dapp} values of G-clamp-N₃ to hsa-mir-4520-1 and its mutants through surface plasmon resonance (SPR) analysis as representative data for RNA-binding of G-clamp. Consequently, we have observed the same binding tendency as the values obtained by fluorescence titration experiments, although the values slightly increased. We have added the following sentences and Figure (new Figure S6).

Result - Large-scale interaction analysis of G-clamp-N₃ with library-1, 2nd paragraph

“Surface plasmon resonance (SPR) analysis also showed the same binding tendency as the values obtained by fluorescence titration experiments, although the values slightly increased (Figure S6).

While the wild-type and mutG2A exhibited strong binding ($K_{Dapp} = 0.10 \pm 0.02$ and $0.044 \pm 0.008 \mu\text{M}$, respectively), mutG7A exhibited much weaker binding ($K_{Dapp} > 50 \mu\text{M}$).”

2-3: Docking studies are helpful in formulating ideas, but more is needed to explain how the bound structure is formed. Are there any other possible bound structures? How are stable structures produced with other mutated RNA sequences?

[Answer-2-3] Thank you for the important comment. We have added the following sentences to explain how the complex structures are formed.

METHODS - Molecular modeling -

“The RNA 3D structures were predicted using RNAComposer. The energy minimization of complex structures between RNA and G-clamp–N₃ was performed using MacroModel after setting G-clamp–N₃ to face the G base so that hydrogen bonds could be formed. OPLS3e and water were used as the force field and solvent, respectively.”

We have added the figures (new Figures S7-S8) to show other possible bound structures. No clear stacking interaction between G-clamp and RNA could be observed, except for the main target (7G of hsa-mir-4520-1). We have also changed the modelling structure of Figure 3 to clearly show the interactions between G-clamp and RNA.

.....

Reviewer 3:

We would like to thank reviewer 3 for the useful comments and suggestions. These have helped us to make considerable improvements to our revised manuscript. We have revised the structure of the figures based on comments and added explanations in the text and legends. We hope that these revisions will be satisfactory.

3-1: Line 33. “small molecule-binding” is more appropriate than “RNA-binding”

[Answer-3-1] Thank you for the suggestion. We have revised it.

3-2: Line 89 through 94 should be in the methods.

[Answer-3-2] Thank you for the suggestion. We have moved the sentences about synthesis into the methods.

3-3: When discussing the analysis of g-clamp interactions starting at line 96, it would be nice to

have a brief outline of RNA ranking in the main text (something simple like what is in the figure caption). It would also be great to know which RNA folding algorithm was used or where the structures came from.

[Answer-3-3] Thank you for the suggestion. We have added the following sentences.

Result - Large-scale interaction analysis of G-clamp-N₃ with library-1, 1st paragraph

“In ranking list S1, the sequences, binding scores, Z-scores, and CVs are shown in order of rank. To understand the binding properties of G-clamp, the numbers of bases in the single-stranded (ss) and double-stranded (ds) RNA regions were investigated using the secondary structures of the pre-miRNA loops predicted by RNAsubopt in the ViennaRNA package⁵¹ (Figure 2). The ssRNA region refers to the terminal loop, bulge, or internal loop. Boxes were generated for each of the five subpopulations based on their rankings.”

3-4: Lines 114-115, why are the only RNA motifs with a common stem used for K_d calculations.

[Answer-3-4] Thank you for the question. We used the RNA motifs with a common stem for K_{Dapp} calculations to keep the motif structure stable. We were concerned that the motif structure would change without a common stem. We have added the following sentence.

Result - Large-scale interaction analysis of G-clamp-N₃ with library-1, 2nd paragraph

“To shorten the common stem and keep the RNA motif structures stable in the titration assays, a shorter common stem (three base pairs), was attached to the motifs (5'-AGC-motif-GCU-3').”

3-5: Line 146, why was Library 1 not used for TO and TO-3? Are the pre-miRNA loops in the Library 2 to the same as those used in Library 1?

[Answer-3-5] Thank you for the questions. When we started the synthesis of TO and TO-3 analogs, the COVID-19 pandemic just started. We thought that creating the virus RNA-binding profiles of these fluorescent indicators would be useful; thus, we added SARS-CoV2 and influenza A virus RNAs to the original Library 1. At this time, four more pre-miRNAs were added to Library-1. The analysis of Library-2 in Figure 4 used the same pre-miRNAs used in Library-1.

3-6: Line 175 and 183, why were viral RNAs only analyzed with TO-PRO derivatives?

[Answer-3-6] Thank you for the question. We have measured K_{Dapp} values using viral RNAs and TO-N₃ or TO-3-N₃ and added the data to Figures 5 and S11-12.

3-7: Lines 194-213. The structure of this section is a bit confusing. The figure panels are not mentioned in order so maybe reorganize the figure or reorganize the main text so there is better flow. If the main text is left as is, maybe have the structures as A, the S/B plots as B, the FID assay as C, and the compounds and tables as D or D-G in the figure.

[Answer-3-7] Thank you for the suggestions. Based on the suggestions, we made the new Figure 6. We separated the FID assay validation data from Figure 6 and created a new figure (Figure 7).

3-8: Lines 226-238 are a bit confusing to me. This section kind of feels out of place compared to the rest of the paper. The AS spectrum fits better than the Che spectra as it relates more to the potential development of new probes for these kind of FID experiments.

[Answer-3-8] Thank you for the comment. As mentioned, the Che spectra did not fit compared to the rest of the paper. Thus, we have removed the discussion and figure of Che spectra.

3-9: For the structural preference analysis, why was RNAsubopt used instead of the normal MFE fold from RNAfold?

[Answer-3-9] Thank you for the question. To accurately reflect the dynamic nature of RNA structures, we employed RNAsubopt to comprehensively assess potential RNA conformations beyond the MFE structure. This methodology, validated by studies such as those by Dominguez D. *et al.* (ref71: Molecular Cell, 2018) and Komatsu K. R. *et al.* (ref30: Nature Communications, 2020), offers a robust framework for understanding the structural preferences of RNA binders.

3-10: For FID assay calculations, why was a reduction of less than a standard deviation used and normalized values greater than 1.5 excluded?

[Answer-3-10] Thank you for the questions. A standard deviation was used to examine whether the change in fluorescence was statistically significant. We made a mistake in METHODS in the original paper, and the correct information is as follows.

“Hits were selected based on a reduction of TO-PRO-1 or TO-PRO-3 signal by less than **two** standard deviations (2σ) from the mean.”

Values greater than 1.5 were excluded because the increase was likely due to compound-derived fluorescence and was considered abnormal.

We have added the following sentence to the main text.

Result - Screening of the novel RNA-binding molecules by fluorescent indicator displacement assay using TO-PRO-1 and TO-PRO-3, 1st paragraph

“We defined the hit threshold as the mean subtracted by twice standard deviations (mean – 2σ).”

3-11: Figure 1: Good flow to the figure. Pretty understandable without the caption, however, a bit of additional information would be nice in the caption.

[Answer-3-11] Thank you for the suggestion. We have added the following sentences to the caption of Figure 1a.

Figure 1 (a)

“(a) Schematic of the large-scale analysis of small molecule–RNA interactions. The RNA structured library consists of an RNA structure region, a common stabilizing stem region, and a barcode region. The 3' end is modified with a fluorescent group. The RNA structure region has 1824 kinds of structure consisting of pre-miRNA loops and repetitive sequences (Library-1). Library-2 contains library-1 plus SARS-CoV-2 and influenza A viral RNAs. The designed RNA structure library was used for the multiplexed pull-down assay with a small molecule immobilized on streptavidin-coated magnetic beads. The enriched RNA structures were analyzed based on the differences in fluorescence intensity observed on DNA barcode microarrays, and the interactions between small molecules and RNA were quantified.”

3-12: Figure 4: I am thinking the definition of hairpin here is referring to the terminal loop. Clarification of this would be great.

[Answer-3-12] Thank you for the suggestion. We have changed hairpin to terminal loop.

3-13: Figure 5: I would suggest breaking this figure up and give each plot a different letter so a direct reference to each plot can be made in the text. I would also keep all the colors the same between A and B.

[Answer-3-13] Thank you for the suggestions. We have changed Figure 5 based on the suggestions.

3-14: Figure 6: The caption does not explain this figure very well. Similar to figure 5, maybe the tables need individual letters so they can be referenced directly. The caption needs to be more descriptive as to what the numbers in the table actually represent. In panel b, there is a missing hyphen in TO-3-N3-2.

[Answer-3-14] Thank you for the suggestions. We have changed Figure 6 based on the suggestions.

REVIEWERS' COMMENTS:

Reviewer #1 (Remarks to the Author):

The authors made efforts to answer all referees comments quite satisfactorily.

Reviewer #2 (Remarks to the Author):

The authors have adequately addressed most of my concerns in the revised manuscript. SPR experiments clearly showed the binding tendency is consistent with the observed results by fluorescence titration experiments. However, this reviewer strongly contends with the explanation regarding the observation of five or more unpaired Gs in high-ranking RNAs.

In the case of mir-6847, ranked second and possessing six unpaired guanines, a critical inquiry arises: Can these six guanines form a G4 structure, and can G-clamps effectively stack onto the G4?

It is reasonable to assume that an increased number of guanines not engaged in hydrogen bonding would enhance the probability of G-clamp binding to RNA containing free guanine. Consequently, the presence of five or more unpaired Gs in high-ranking RNAs is primarily attributed to the heightened likelihood of G-clamp binding, while binding to G4 structures is less probable.

If the authors intend to maintain the current explanation of G-clamp-binding to G-rich RNA in their library, they must provide supporting evidence. Presently, the explanation remains speculative without substantial empirical support.

Reviewer #3 (Remarks to the Author):

The authors have addressed all my comments and I believe this is ready for publication.

Point-by-point responses to the reviewers' comments (2)

Reviewer 1:

We would like to thank reviewer 1 for checking our revised manuscript.

Reviewer 2:

We would like to thank reviewer 2 for checking our revised manuscript and providing helpful comments and suggestions. In light of reviewer #2 concerns, we have reconsidered the G-clamp selectivity description and removed the following sentence:

"RNAs with five or more unpaired Gs are particularly enriched in Figure S3 because G-clamp binds not only specific unpaired Gs but also G4 or G4-like structures, and such structures are often formed when multiple Gs exist in a single-stranded region."

As reviewer #2 pointed out, this explanation was speculative.

We hope that this revision will be to his/her satisfaction.

The authors have adequately addressed most of my concerns in the revised manuscript. SPR experiments clearly showed the binding tendency is consistent with the observed results by fluorescence titration experiments. However, this reviewer strongly contends with the explanation regarding the observation of five or more unpaired Gs in high-ranking RNAs.

In the case of mir-6847, ranked second and possessing six unpaired guanines, a critical inquiry arises: Can these six guanines form a G4 structure, and can G-clamps effectively stack onto the G4?

It is reasonable to assume that an increased number of guanines not engaged in hydrogen bonding would enhance the probability of G-clamp binding to RNA containing free guanine. Consequently, the presence of five or more unpaired Gs in high-ranking RNAs is primarily attributed to the heightened likelihood of G-clamp binding, while binding to G4 structures is less probable.

If the authors intend to maintain the current explanation of G-clamp-binding to G-rich RNA in their library, they must provide supporting evidence. Presently, the explanation remains speculative without substantial empirical support.

Reviewer 3:

We would like to thank reviewer 3 for checking our revised manuscript.